# Optical-field driven charge-transfer modulations near composite nanostructures

Kwang Jin Lee [1,2,3 ✉], Elke Beyreuther [4], Sohail A. Jalil[1,5], Sang Jun Kim [6], Lukas M. Eng[4], Chunlei Guo [1 ✉] & Pascal André [3,7,8 ✉]

Optical activation of material properties illustrates the potentials held by tuning light-matter interactions with impacts ranging from basic science to technological applications. Here, we demonstrate for the first time that composite nanostructures providing nonlocal environments can be engineered to optically trigger photoinduced charge-transfer-dynamic modulations in the solid state. The nanostructures explored herein lead to out-of-phase behavior between charge separation and recombination dynamics, along with linear charge-transfer-dynamic variations with the optical-field intensity. Using transient absorption spectroscopy, up to 270% increase in charge separation rate is obtained in organic semiconductor thin films. We provide evidence that composite nanostructures allow for surface photovoltages to be created, which kinetics vary with the composite architecture and last beyond optical pulse temporal characteristics. Furthermore, by generalizing Marcus theory framework, we explain why charge-transfer-dynamic modulations can only be unveiled when optic-field effects are enhanced by nonlocal image-dipole interactions. Our demonstration, that composite nanostructures can be designed to take advantage of optical fields for tuneable charge-transfer-dynamic remote actuators, opens the path for their use in practical applications ranging from photochemistry to optoelectronics.

[1] The Institute of Optics, University of Rochester, Rochester, New York, USA. [2] Department of Physics, Ewha Womans University, Seoul, South Korea. [3] CNRS-Ewha International Research Center, Ewha Womans University, Seoul, South Korea. [4] Institut für Angewandte Physik, Technische Universität Dresden, Dresden, Germany. [5] Changchun Institute of Optics, Fine Mechanics, and Physics, Changchun, China. [6] Ellipso Technology Co. Ltd, Suwon, South Korea. [7] Laboratoire des Multimatériaux et Interfaces, Université Claude Bernard Lyon 1, UMR CNRS 5615, Villeurbanne, France. [8] RIKEN, Wako, Japan. ✉email: klee102@ur.rochester.edu; guo@optics.rochester.edu; pjpandre@alum.riken.jp

Tuning, activating, and enhancing specific properties are among the most common challenges undertaken by scientists. To reach this goal, interface and heterostructure nanoengineerings are attractive strategies applied for energy transduction enhancement[1], nanophotonic circuitry design[2,3], quantum information processing[4], catalysis activation[5,6], as well as tunable narrowband photo-detection[7] and scattering[8]. Triggers include position[8], strain[9], gating[10], charge injection[11–13], along light power[13,14], wavelength[15,16], or polarization[17], with optical approaches being of unique interest for remote actuation.

Anisotropic to flat nanostructures can be designed with materials ranging from 2D to alternating layers of metal, dielectric, or semiconductor of selected permittivity[18–24]. Metal-oxide multilayer composite nanostructures provide non-local environments, which are attracting attention[25], as they were shown to affect luminescence[26–30], photodegradation[31], and wetting properties[32], as well as to slow down charge transfer dynamics in donor:acceptor systems[33]. Charge transfer processes are fundamental steps impacting on biology and photosynthesis[34], chemistry[35,36], as well as electronics[37,38]. However, remotely modulating charge-transfer dynamics in the solid state remains challenging despite fundamental and technological interest. Actively controlling the time scale over which charge separation and recombination occur is critical especially in the solid state, which is relevant to most optoelectronic devices and circuitry. To open the path toward light-based technologies with remotely tunable charge-transfer dynamics, the underlaying cause of relevant phenomena[25–32] need to be both identified and rationalized.

To address this challenge, we used non-invasive time-resolved spectroscopies to probe a range of ultra-fast phenomena. Transient absorption allows monitoring of charge-transfer dynamics between donors and acceptors[26–29,33,36,38–44]. Surface photovoltage measurements provide insights on the photon-induced electrical-potential change across interfaces[45,46]. It is a well-established and powerful technique involved for instance in recent investigations of charge separation which can occur in inorganic structures[22]. Organic semiconductors were selected because of their relevance to solution-processable electronics[43,44,47–52], and the adjustability of the relative position of Donor:Acceptor groups, in terms of energy levels and spatial positions. The latter characteristic makes organic donor:acceptor dyads attractive model systems to investigate charge separation and recombination mechanisms due to reduced domain sizes and donor:acceptor interface distributions, which otherwise also impact on exciton formation, diffusion, dissociation, and recombination. The donor:acceptor molecules were deposited on top of metal/oxide multilayer structures with selected thicknesses of the top oxide material.

In a previous and related work[33], we focused on the number of pairs of composite nanostructures and inserted image–dipole interactions in Marcus theory framework to show that both charge separation and recombination slowed down with the number of metal-dielectric pairs. Albeit exciting in itself, the limitations of this earlier work included, however, to consider the slowdown of charge-transfer-state dynamics as only controlled by one substrate-structural parameter, it could not identify ways to impact differently separation and recombination characteristic times and only the driving force seemed affected. In the present work, we address these limitations by focusing on 4-pair composite nanostructures, and we use the thickness of the top Al$_2$O$_3$ spacer to reveal unexpected optical-field effects. We show that engineered composite nanostructures provide optical control over charge transfer dynamics in the solid state. On such nanostructures, we evidence that charge separation and recombination dynamics display out-of-phase modulations, one increasing when the other decreases. This so-far unreported behavior is shown to result from a long-lasting optical-field effect mediated by surface photovoltages, an overlooked parameter in multilayered nanostructures, herein enhanced by nonlocal image-dipole interactions. Out-of-phase charge-transfer-dynamic modulations are successfully described after introducing optical-field intensity in a generalized Marcus theory framework. This theoretical contribution reproduces the trend of the experimental data and it also shows that both driving force and reorganization energy are affected.

The present results shine lights on the potentials offered by nonlocal environments and by composite nanostructures engineering. We show how charge-transfer dynamics, essentially governed by thermodynamic mechanisms, can be controlled in the solid state by electromagnetic fields acting as remote actuators with differentiated impacts on charge separation and charge recombination. The effect is coined as nonlocal enhanced optical field (NEOF) and this discovery bears wide potentials ranging from basic science to nanophotonics.

## Results

**Sample configurations and charge-transfer-dynamic modulations.** Figure 1a presents the schematic of the composite nanostructures and the organic semiconductor used in this work. The composite nanostructures are made of 4 pairs (4p) of 10-nm-thick silver (Ag) and aluminum oxide (Al$_2$O$_3$) successive layers, with the last Al$_2$O$_3$ top-cover ranging from 10 nm to 1 μm. The donor:acceptor molecule is made of triphenylene and perylene diimide moieties chemically grafted with a flexible decyloxy bridge leading to a dyad, which can be spin-coated on top of the composite nanostructures[41,53].

The sub-picosecond transient absorption spectra presented in Fig. 1b, c are generated by pumping at 325 nm, and monitoring at 725 nm with constant time intervals. The relative transmittance variation ($\Delta T/T$) curves are obtained from annealed dyad thin films on composite nanostructures for short and long delay times, respectively. They display typical transient absorption features resulting from the formation and disappearance of photoinduced charge-transfer states. In Fig. 1b, the $\Delta T/T$ signal varies progressively as acceptor radical anions are formed, corresponding to photoinduced charge separation (CS) and leading to the formation of charge-transfer states. These are equivalently described as dipole moments between triphenylene and perylene diimide charged units. A plateau is reached within 3 ps, when no more charge-transfer states induced by the pump pulse are formed. Over a much longer times scale and for longer time interval, the signal recovery displayed in Fig. 1c monitors charge recombination (CR). It corresponds to the progressive disappearance of charge-transfer states, until a plateau is reached within 700 ps. The insets show that both dynamics are properly described with single exponential characteristic times. The kinetics obtained on a range of composite nanostructures are presented in Table 1.

As recently reported, charge-transfer-dynamic slowdown results from nonlocal image-dipole-interaction effects supported by metal-dielectric multilayers with charge separation and recombination within the dyad film being in normal and inverted Marcus regions, respectively[33]. All the 4p-composite nanostructures used herein lead to slower charge-transfer dynamics than measured on plain fused silica, which charge-transfer-characteristic times and uncertainties are illustrated in Fig. 1d, e by the horizontal straight lines. Charge separation (CS) is the fastest for 10-nm and 1-μm dielectric composite nanostructures (Fig. 1d), whereas charge recombination (CR) is the slowest for 10-nm top-cover composite nanostructures (Fig. 1e). Interestingly, charge transfer dynamics present a non-monotonous,

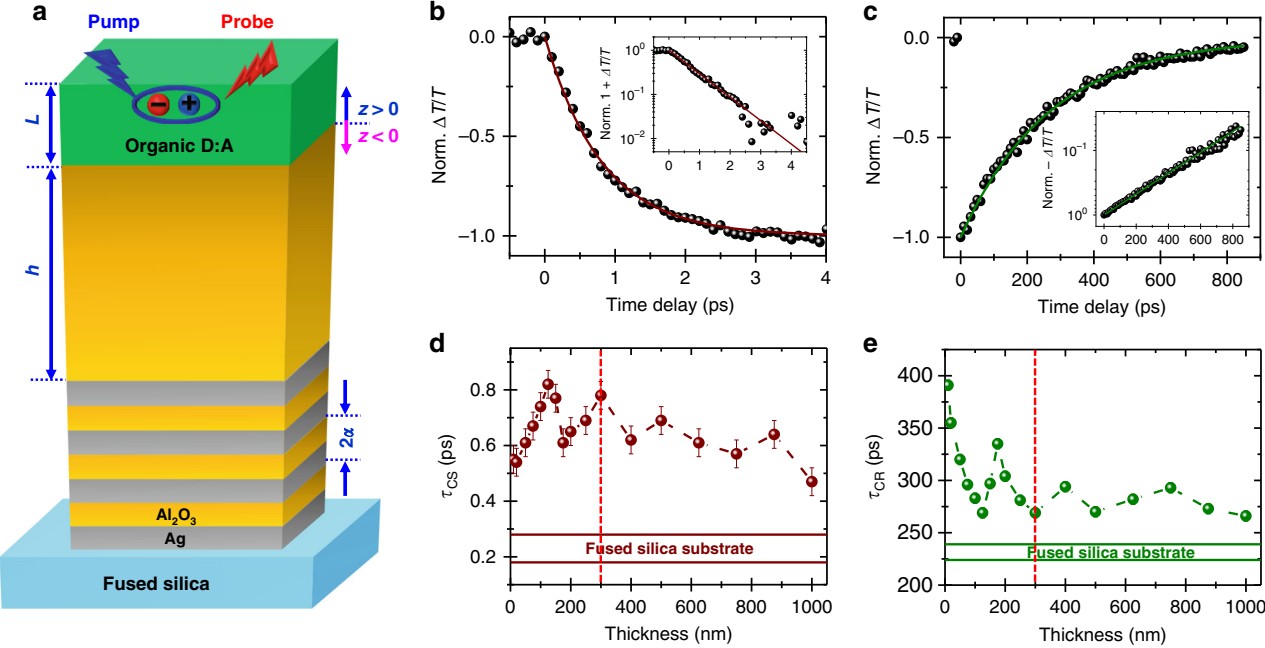

**Fig. 1 Composite nanostructure and transient absorption data. a** Schematic representation of the Ag-Al$_2$O$_3$ composite nanostructure covered by an organic dyad film with their characteristic parameters: number of pairs ($p$), pair thickness and periodicity ($2a = 20$ nm), metal volume fraction ($f_v = 0.5$), Al$_2$O$_3$ top-thickness ($h$), distance from the Al$_2$O$_3$-Donor:Acceptor (D:A) molecules interface ($z$), and dyad-film thickness ($L=40$ nm). **b, c** Normalized relative transmittance variation for charge separation and charge recombination, respectively, as a function of the time delay with a 125-nm-thick Al$_2$O$_3$ composite nanostructure top-cover (inset: semi-log scale of $1+\Delta T/T$, and $-\Delta T/T$, note that to illustrate the signal variation the scales in the inset are inverted), respectively. Colored lines are exponential fits. **d, e** Charge separation (CS) and charge recombination (CR) dynamics as a function of the thickness of the Al$_2$O$_3$ composite nanostructure top-cover; the horizontal straight lines correspond to the charge transfer dynamics and their uncertainties measured on reference fused silica substrates. All the measurements were completed in transmittance mode, with a 325-nm photo-excitation and a 725 nm-probe beam.

**Table. 1 Characteristic times.**

| Substrate | $\tau_{CS}$ (ps)[a] | $\tau_{CR}$ (ps)[b] | $\tau_{CS}/\tau_{CR}$ (x10$^{-3}$) |
|---|---|---|---|
| Fused Silica | 0.23 ± 0.05 | 226 ± 5 | 1.0 ± 0.2 |
| $h = 10$ nm | 0.55 ± 0.05 | 391 ± 5 | 1.4 ± 0.2 |
| $h = 20$ nm | 0.54 ± 0.05 | 355 ± 5 | 1.9 ± 0.2 |
| $h = 50$ nm | 0.61 ± 0.05 | 320 ± 5 | 2.3 ± 0.2 |
| $h = 75$ nm | 0.67 ± 0.05 | 296 ± 5 | 2.6 ± 0.2 |
| $h = 100$ nm | 0.74 ± 0.05 | 283 ± 5 | 3.1 ± 0.3 |
| $h = 125$ nm | 0.85 ± 0.05 | 272 ± 5 | 2.6 ± 0.2 |
| $h = 150$ nm | 0.77 ± 0.05 | 297 ± 5 | 1.8 ± 0.2 |
| $h = 175$ nm | 0.61 ± 0.05 | 336 ± 5 | 2.1 ± 0.2 |
| $h = 200$ nm | 0.65 ± 0.05 | 304 ± 5 | 2.5 ± 0.2 |
| $h = 250$ nm | 0.70 ± 0.05 | 281 ± 5 | 2.9 ± 0.3 |
| $h = 300$ nm | 0.78 ± 0.05 | 269 ± 5 | 2.1 ± 0.2 |
| $h = 400$ nm | 0.62 ± 0.05 | 294 ± 5 | 2.6 ± 0.2 |
| $h = 500$ nm | 0.69 ± 0.05 | 270 ± 5 | 2.2 ± 0.2 |
| $h = 625$ nm | 0.61 ± 0.05 | 282 ± 5 | 2.0 ± 0.2 |
| $h = 750$ nm | 0.57 ± 0.05 | 293 ± 5 | 2.4 ± 0.2 |
| $h = 875$ nm | 0.64 ± 0.05 | 272 ± 5 | 1.8 ± 0.2 |

Charge separation ($\tau_{CS}$), recombination ($\tau_{CR}$), and their ratio ($\tau_{CS}/\tau_{CR}$) obtained by transmittance mode transient absorption spectroscopy of donor:acceptor thin films on 4 pair substrates of different spacer thickness ($h$).
[a]100 fs.
[b]10 ps time interval.

almost periodic, variation with the thickness of the composite nanostructures top-cover. This is unexpected as the image-dipole-interaction description suggests a continuous decay of the nonlocal effect with the thickness of the dielectric. Noticeably, charge separation and recombination characteristic-time variations show an out-of-phase relationship. The peaks and valleys of

$\tau_{CS}$ and $\tau_{CR}$ are inversed, for instance, ~125, 175, and 300 nm, with the latter, materialized by red dashed vertical lines, but do not compensate one another, as displayed in Fig. 2a. Noticeably, the modulation amplitude decreases as the donor:acceptor film is further away from the nearest metal–Al$_2$O$_3$ interface (Fig. 1d and e, Fig. 2a). Overall, when compared to fused-silica data, charge separation, and recombination on top of composite nanostructures varied by a maximum factor of 3.7 and 1.7, respectively. They correspond to spacer thicknesses of 125 nm and 10 nm, with the ratios of 0.85 ps to 0.23 ps and 391 ps to 226 ps, as shown in Table 1. When comparisons are completed between kinetics obtained on composite nanostructures, these factors reach ~1.57 and ~1.44, corresponding to relative variations of 57% and 44%, respectively. This charge-separation-maximum variation is obtained between spacer thicknesses of 125 nm (0.85 ps) and 10 nm (0.55 ps), while this charge-recombination maximum is observed between spacer thicknesses of 10 nm (391 ps) and for 300 nm (269 ps).

Importantly, control experiments in transmittance and reflectance show that charge-transfer-dynamics data are measurement mode independent. On 200 nm thick Ag composite nanostructures, charge separation appears to be constant within the experimental uncertainty, whilst charge-recombination oscillations with the Al$_2$O$_3$ thicknesses are slightly larger than the experimental uncertainty. They are also much smaller (~8%) than those observed with 4p-composite nanostructures (~44%). In addition, the charge-transfer-dynamic modulation periodicity differs on thick Ag film and 4p-composite nanostructures, from ~250 nm to ~150 nm, respectively. Consequently, experimental and fit uncertainties, measurement mode artefacts, charge transfer to the metal layers and simple image-dipole-interactions can be ruled out to explain these out-of-phase

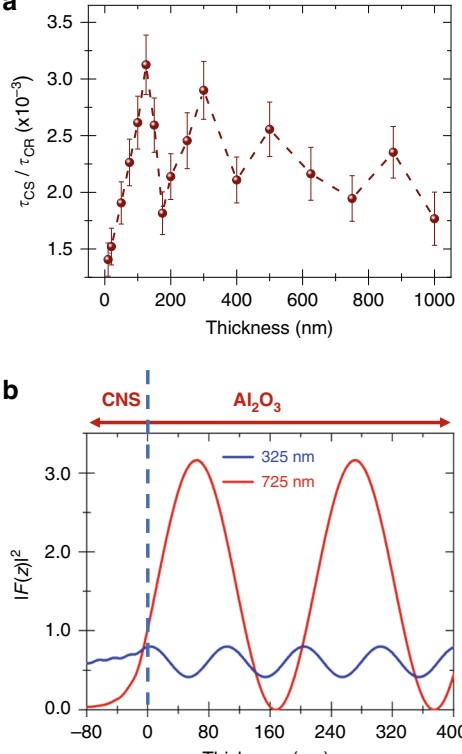

**Fig. 2 Analysis of charge-transfer dynamic and optical field in composite nanostructures. a** Variations of charge separation (CS) and charge recombination (CR) ratio as a function of the $Al_2O_3$ top-layer thickness. **b** Optical-field intensity distributions of both pump (325 nm) and probe (725 nm) beams across the $Al_2O_3$ top-covers.

charge-transfer-dynamic modulations, which appear to be a specific, but non-trivial, effect of composite nanostructures. At least because the present organic-semiconductor charge transfers are non-emissive, we can also already conclude that the herein reported phenomenon is unrelated to Purcell effects[54], which we have previously studied on composite nanostructures[29].

However, looking at the optical beam going through the sample, it appears that the probe beam outweighs the impact of the pump beam as displayed in Fig. 2b. We explore further this feature in the following section.

**Optical-field impact on charge-transfer dynamics**. These photoinduced charge-transfer-dynamics oscillations are reminders of interference effects on fluorescence lifetime, quantum efficiency and energy-transfer modulations near metal interfaces[39,55,56]. However, we stress that these are related to radiative dipoles where the change of radiative decay rate is induced by interferences between the direct radiation of an emitter and the same radiation reflected by the metal surface. However, in the present work, the photoluminescence is quenched and the monitored non-radiative charge transfers are of a different nature. Indeed, how a metal surface affects transition (radiative) dipoles is described through electromagnetic theory, which formalism considers the standing fields and surface reflectivity explicitly.

In contrast, charge-transfer dynamic is essentially governed by thermodynamics, in which electro-magnetic field effect is usually absent. Other examples of the impact of interferences include solar cells, where ideally photoluminescence is also quenched. Photo-conversion efficiencies are related to the time average of the energy dissipated per second, which depends on the locally absorbed energy and is associated with the interferences between

incident light and its reflection on the metal electrodes[43]. However, the effect is related to the number of excitons formed and not to charge-transfer dynamics.

To explore standing-wave contributions, we calculated the optical-field-intensity distribution, $|F|^2$, i.e., the light intensity, in the Donor:Acceptor films and the composite nanostructures using invariant imbedding method[29,57,58] and optical constants measured by ellipsometry. Figure 3a–c present $|F|^2$ as a function of the wavelength for three substrates. $|F|^2$ calculated on fused silica is relatively small and homogeneous compared to the composite nanostructures, which present much larger $|F|^2$ values at 725 nm than at 325 nm, the probe and the pump wavelength respectively. $|F|^2$ cross-sections are presented in Fig. 3d–f. Whereas $|F_{725nm}|^2$ is transmitted through plain fused silica (Fig. 3d), the large reflectivity of the metal-dielectric stack leads $|F_{725nm}|^2$ to tail off in the composite nanostructure. The optical field intensity is very large inside the Donor:Acceptor film on top of the 10-nm-thick $Al_2O_3$ cover composite nanostructure (Fig. 3e), with its maximum value reached at the Donor:Acceptor-air interface. In contrast, Fig. 3f shows that with the 100-nm-thick $Al_2O_3$ spacer-composite-nanostructure, $|F_{725nm}|^2$ is reduced by a factor of ~2 and its maximum value is located inside the dielectric spacer. These calculations suggest that optical-field intensities within the nanostructures and organic semiconductor thin film can be tuned by changing the thickness of the top $Al_2O_3$ layer. In the present situation, where there is no emission from the organic semiconductor molecules, the calculated variations of the optical fields could be linked directly or indirectly with the oscillatory phenomena reported in Fig. 1. They could also become an attractive tool to qualitatively model the charge-transfer-dynamic modulations herein reported, even though at the likely identifying a bridge between intensity and kinetics. This is what the following sections explore.

**Evidence of long-lasting surface photovoltages**. In pump-probe transient experiments, the light field is only present when the pump and probe short pulses are passing through the sample. During the time delay between the pulses, i.e., when the charge transfers occur, there should not be any light. Consequently, for any optical effect to play a role on composite-nanostructure's charge-transfer dynamics, it should outlast the 60 fs pulse width and the 5 kHz repetition-rate-time scale of the incident-transient-absorption-optical-beam parameters, as well as the few hundred picoseconds associated with the charge-recombination-time scale. As discussed above, an incident-optical field can provide the oscillatory behavior, but the dynamics of the optical beam used in the transient-absorption experiments are so fast that it then needs to trigger a build-up, which will impact charge-transfer dynamics long after the incident waves have vanished from the nanostructures. The build-up phenomenon ought to work for a much longer time scale than those mentioned above. Combining the charge-transfer-dynamic oscillations with the requirement to outlast the incident waves and charge-transfer-characteristic times, points toward an indirect optical effect mediated by the architecture of the composite nanostructures - a mediation which has not been considered before. In this context, surface photovoltages are exceptionally relevant as they are photoinduced and exhibit – at least in a number of wider-gap materials – much longer characteristic times than those of the incident waves. The validity of this hypothesis was demonstrated by completing Kelvin probe time-resolved surface photovoltage measurements on three bare exemplary composite nanostructures.

In principle, interfaces, which exhibit an optically accessible depletion zone due to an energy band bending can show a change in the electrical potential upon illumination due to charge

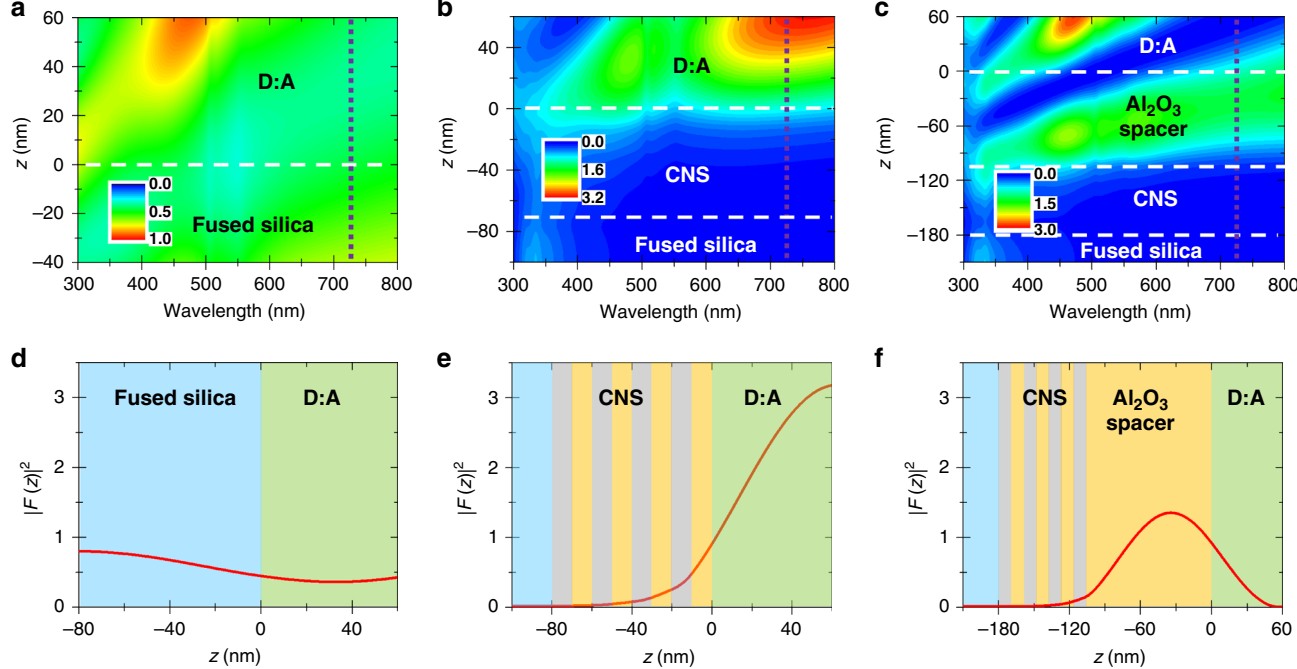

**Fig. 3 Optical modulation induced by the substrate structure. a–c** False-color plot of the optical field intensity, $|F^2|$, as a function of incident wavelengths and position within the sample. **d–f** Optical-field intensity distribution within the sample corresponding to a cross section at 725 nm of **a–c** with the background colors representing the sample composition. Fused silica substrate (**a, d**), 4p-composite nanostructure (CNS) with an $Al_2O_3$ top layer of 10 nm (**b, e**), and 100 nm (**c, f**), respectively, and Donor:Acceptor (D:A) thin film. The horizontal lines materialize the different layers of materials.

exchange processes between the bands or between trap states and the bands, which is referred to as surface photovoltage[46]. This can be expected for the $Ag/Al_2O_3$ interfaces and the topmost $Al_2O_3$ surface (see Supplementary Fig. 3 for a schematic illustration of a tentative band diagram of the present system). Indeed, the three test structures exhibit a surface photovoltage, as presented in Fig. 4. No measurable surface photovoltage signal could be extracted from a single Ag layer on fused silica substrates, showing that the metal-dielectric multilayer structure is essential for the appearance of surface photovoltage signals.

Figure 4a, b display typical surface-photovoltage transients upon ON and OFF switching of a 633-nm laser beam with a photon flux comparable to the transient absorption experiments. In the most complex cases, surface-photovoltage transients can only be numerically or partially analytically fitted[59]. However, biexponential fits were suitable for all the data obtained with these systems, as illustrated in the insets. Importantly, both amplitude and characteristic time constants vary with the structure of the composite as shown by the fitted curves in Fig. 4c, d. In Fig. 4e, the surface photovoltage signal at saturation is displayed as a function of the thickness of the last $Al_2O_3$ layer of the composite nanostructures and against the photon flux inside the structure (i.e., the quantity 100-R (%), with R being the reflectance standing for the intensity not being reflected but passing the sample or being absorbed). It confirms, as expected, that both are linked to one another. Figure 4f shows that both fast and slow components of the surface photovoltage dynamics are affected by the composite nanostructures in a non-monotonous manner. The rise measured in bright mode is systematically faster than the recovery obtained in dark mode, with characteristic times ranging from seconds to tens of minutes. In either case, the surface photovoltage supported by the composite nanostructures clearly outlasts the repetition rate of the present transient-absorption measurements. These surface photovoltages, which likely origin from charge exchange between in-gap states within the $Al_2O_3$ bandgap and one of the bands, arise under both pulsed and

continuous illumination. Therefore, the unexpected long persistence of optical field effect can be linked to surface photovoltages stabilized by the image-dipole interactions supported by the composite nanostructures.

To the best of our knowledge, such surface photovoltages have not been reported in the context of composite nanostructures so far, and it is also possible that they contribute to responses of materials deposited on multilayer structures including plasmonic and meta-materials/-surfaces. In the present study, long lasting surface photovoltages contribute to optical effects which are supported by the composite nanostructures and combine to image-dipole interactions on charge-transfer states to form nonlocal enhanced optical field (NEOF) resulting in the charge-transfer-dynamic modulations. These should naturally be described in the framework of Marcus theory, as presented in the following section.

**Generalized Marcus theory.** In a dielectric continuum, nonadiabatic charge transfer (CT) reaction rates, $k_{CT}$, can be described with Marcus theory in terms of reorganization energy ($\lambda_{CT}$), thermal energy ($k_BT$), electronic coupling between the initial and final states ($V_{DA}$, charge transfer integral), activation Gibbs free energy ($\Delta G_{CT}^*$) and Gibbs free energy gain ($\Delta G_{CT}$, driving force) as stated in Eqs. (1) and (2)[33,40]:

$$k_{CT} = \left(\frac{4\pi^3}{h^2\lambda_{CT}k_BT}\right)^{1/2}|V_{DA}|^2\exp\left(-\frac{\Delta G_{CT}^*}{k_BT}\right) \quad (1)$$

$$\Delta G_{CT}^* = \frac{(\lambda_{CT}+\Delta G_{CT})^2}{4\lambda_{CT}} \quad (2)$$

For illustration purposes, these are represented in Fig. 5, with the parabolic energy levels of the reactant and the product shifted along the reaction coordinates and energy axis.

The parabolic relation predicts a maximum of $k_{CT}$ at the barrier-less point. In the 'normal' region ($-\Delta G_{CT}<\lambda_{CT}$), a driving

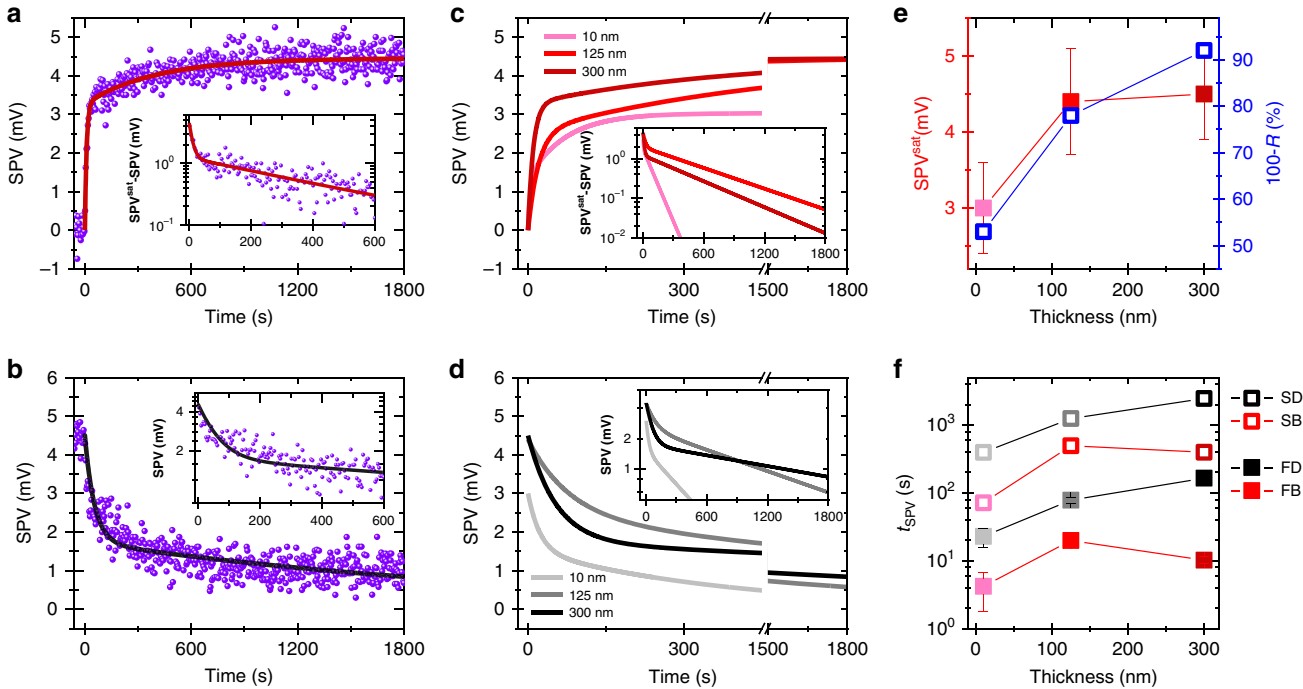

**Fig. 4 Surface photovoltages dynamics and amplitudes supported by composite nanostructures. a, b** Typical increase and decrease of the surface-photovoltage amplitude as a function of time upon ON and OFF switching of a laser beam (inset: semi-log scale of the signal absolute value), respectively. The solid curves are the fits of the rise and decay obtained on a 4p-300 $Al_2O_3$ substrate. **c, d** Fitted surface-photovoltage rise and decay with time of 3 composite nanostructures. Note that no signal could be measured on single Ag thin film deposited on the fused silica substrate. **e** Amplitude of the surface-photovoltage signal and photon flux within the nanostructure (represented by the quantity "100-R (%)" with R being the reflectance) plotted versus the composite nanostructure $Al_2O_3$ top layer thickness. **f** Dynamics of the surface-photovoltage rise and decay under bright (B) and dark (D) conditions with the slow (SD, SB) and fast (FD, FB) component represented by the empty and filled symbols, respectively. All the measurements were completed on bare composite nanostructures, with a 633 nm photo-excitation and at ambient conditions with stabilized temperature and air humidity.

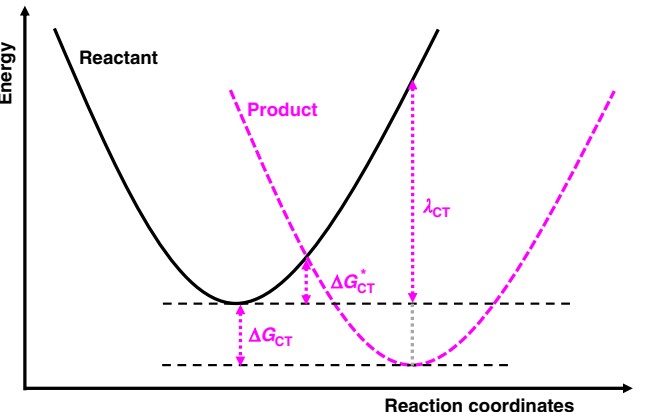

**Fig. 5 Schematic representation of charge transfer.** Energy parabola shift along the energy axis and the reaction coordinates showing the Gibbs free gain ($\Delta G_{CT}$, driving force), the activation energy ($\Delta G_{CT}^*$), and the reorganization energy ($\lambda_{CT}$).

force increase (i.e., more negative $\Delta G_{CT}$) reduces $\Delta G_{CT}^*$, hence increases $k_{CT}$. In the 'inverted' region ($-\Delta G_{CT} > \lambda_{CT}$), increasing the driving force reduces $k_{CT}$ as the activation energy decreases. In the generalized Marcus theory, charge-transfer-state dipoles and their image potentials induce an energy-perturbation term formalized in the image-dipole interaction (IDI) modified permittivity. This approach describes qualitatively well the resulting nonlocal effects on charge-transfer dynamics[33]. However, IDI alone monotonously decreases with distance and then fails at explaining both the modulations and the phase shift of the

charge-transfer dynamics herein reported. By analogy between external electric[60,61] and optical fields, we argue that the external electromagnetic waves induce a perturbation of the driving force and reorganization energy due to the quasi-static dipole moment of charge-transfer states. The mathematical formalism is detailed in the Supplementary Information, but the key parameters are i) the difference between the dipole moment vectors in the initial and final charge transfer states ($\Delta\vec{\mu}$), and ii) the optical field inside the medium ($F$). The perturbations are then:

$$\Delta G_{CT}^{NEOF}(p, F) = \Delta G_{CT}^0 + \delta G_{CT}^{IDI}(p) - \Delta\vec{\mu}.\vec{F} \qquad (3)$$

$$\lambda_{CT}^{NEOF}(F) = \lambda_{CT}^0 - \Delta\vec{\mu}.\vec{F} \qquad (4)$$

where $\Delta G_{CT}^0$ and $\lambda_{CT}^0$ are the intrinsic Gibbs free energy gain and reorganization energy of the charge-transfer (CT) state, respectively, and are independent of both optical fields and image-dipole interactions (IDI). $\delta G_{CT}^{IDI}$ is a perturbation resulting from the charge-transfer-state-dipole-image potentials, and depends on the number of metal-dielectric pairs.

Assuming that the donor excited and ground states do not carry any dipole moment, that $V_{AD}$ is independent of the optical field, and considering that $-\Delta\vec{\mu}.\vec{F}$ is a perturbation, Taylor series expansion to the 2nd order leads to the following expression:

$$k_{CT}^{NEOF} \approx k_{CT}^{IDI}.\left(1 + \delta_{F1}F + \delta_{F2}F^2\right) \qquad (5a)$$

$$\frac{1}{\tau_{CT}^{NEOF}} \approx \frac{1}{\tau_{CT}^{IDI}}.\left(1 - \delta_{F1}F - \delta_{F2}F^2\right) \qquad (5b)$$

with the image-dipole-interaction contribution expressed as

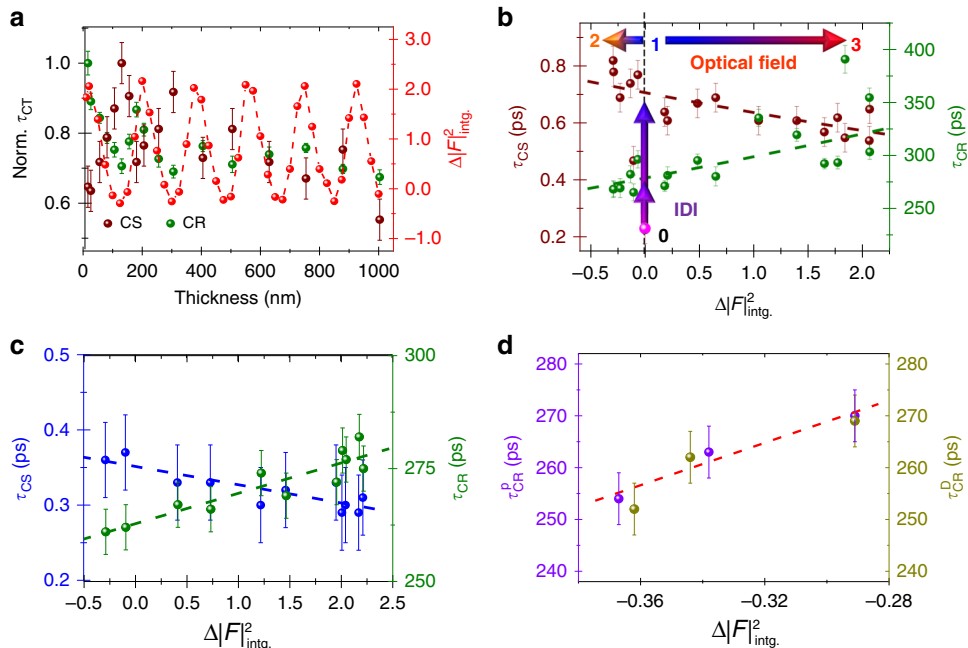

**Fig. 6 Analysis of charge-transfer-dynamic-nanophotonic control by nonlocal enhanced optical-field effects. a** Variations of charge separation (CS, ●) and charge recombination (CR, ●) normalized dynamics and integral of the optical field (●) as a function of the $Al_2O_3$-top-layer thickness. **b** Linear charge separation (CS: ●) and recombination (CR: ●) dynamics dependence with the integral of optical field intensity of a 725-nm-incident beam across the organic Donor:Acceptor thin films on top of 4p-composite nanostructures. The bottom pink data points were obtained on fused silica substrate, the vertical and horizontal arrows materialize nonlocal image-dipole-interaction (IDI) effects and nonlocal enhanced optical-field effects, labeled (0), (1) and (2-3), respectively. **c** Linear charge separation (CS: ●) and recombination (CR: ●) dynamics dependence with the integral of 725 nm optical field intensity across the organic Donor:Acceptor thin films on top of 200-nm-Ag-composite nanostructures. **d** Charge-recombination dynamics plotted against the integral of 725-nm-optical-field intensity for different probe-beam powers (CR-P: ●) and diameters (CR-D: ●) measured on top of Ag-$Al_2O_3$ composite nanostructures covered with 125 nm thick $Al_2O_3$. The dashed lines are a guide.

$$k_{CT}^{IDI} \approx k_{CT}^0 . \exp\left(-\frac{2G_{\lambda-CT}^0 \delta G_{CT}^{IDI} + \delta G_{CT}^{IDI\,2}}{4\lambda_{CT}^0 k_B T}\right) \quad (6)$$

with $G_{\lambda-CT}^0 = \Delta G_{CT}^0 + \lambda_{CT}^0$ and

$$k_{CT}^0 = \left(\frac{4\pi^3}{h^2 \lambda_{CT}^0 k_B T}\right)^{1/2} |V_{DA}|^2 \exp\left(-\frac{G_{\lambda-CT}^0}{4\lambda_{CT}^0 k_B T}\right) \quad (7)$$

Averaging over all the dipole-optical field orientations nullifies the linear term $\langle \delta_{F1} \rangle$ and only the quadratic field contribution remains, corresponding to the optical-field intensity. This analysis is supported by Fig. 6a showing that the variation of the integral of the optical-field intensity across Donor:Acceptor films, $\Delta|F|_{intg.}^2$, presents oscillations of comparable periods as charge-transfer-characteristic times and that qualitatively the $\Delta|F|_{intg.}^2$ maxima are well matched with those of the out-of-phase charge separation (CS) and charge recombination (CR).

Figure 6b presents the experimental charge-transfer-dynamic data as a function of the difference between the composite nanostructures' and fused-silica substrates' optical-field intensities at the probe-beam wavelength. At zero, there is no image-dipole interaction (IDI) contribution, charge-transfer dynamic is the fastest. For charge separation, we obtained 270% enhancement at $\Delta|F|_{intg.}^2 = -0.3$. As expected from the arguments leading to Eq. (5), charge-transfer dynamics on composite nanostructures present almost linear variations with the optical-field intensity. The agreement is better for charge separation than for charge recombination, top and bottom lines in Fig. 6b, respectively.

Noticeably, the slopes are negative with charge separation (CS), positive with charge recombination (CR). This is consistent with charge recombination of the present Donor:Acceptor system being located near the barrierless point[33], where $G_\lambda^0 - CT \approx 0$; then we obtain:

$$\delta_{F2\,\text{Barrierless}} = \left(\frac{3}{4\lambda_{CT}^0} - \frac{1}{k_B T}\right)\frac{(\Delta\mu)^2}{2\lambda_{CT}^0} \quad (8)$$

In this case, $\langle \delta_{F2} \rangle$ is dominated by the thermal energy and is negative at room temperature. Naturally, the inverse of $k_{CR}$, i.e., the characteristic time $\tau_{CR}$, should present a positive slope as observed experimentally. The behavior of charge separation is harder to grasp because $\langle \delta_{F2} \rangle$ equation is more complex. However, insights are gained from the optical-field perturbation induced on the driving force only. The expression of $\langle \delta_{F2} \rangle$ then reads:

$$\delta_{F2\,\text{Driving Force}} = \left(\frac{G_{\lambda-CT}^{IDI\,2}}{2\lambda_{CT}^0 k_B T} - 1\right) \cdot \frac{(\Delta\mu)^2}{8\lambda_{CT}^0 k_B T} \quad (9)$$

where $G_{\lambda-CT}^{IDI} = G_{\lambda-CT}^0 + \delta G_{CT}^{IDI}$.

With this expression, the interplay between optical field and image-dipole-interaction (IDI) effects is revealed. Eq. (9) demonstrates theoretically that the optical-field effect is enhanced by IDI effects. A large IDI contribution leads to large $G_{\lambda-CT}^{IDI}$ and $\delta_{F2}$, which multiplies $F^2$ in Eq. (5). $\langle \delta_{F2} \rangle$ also changes sign near

$$|G_{\lambda-CT}^{IDI}| = \sqrt{2\lambda_{CT} k_B T} \quad (10)$$

implying a positive-negative slope transition when charge-

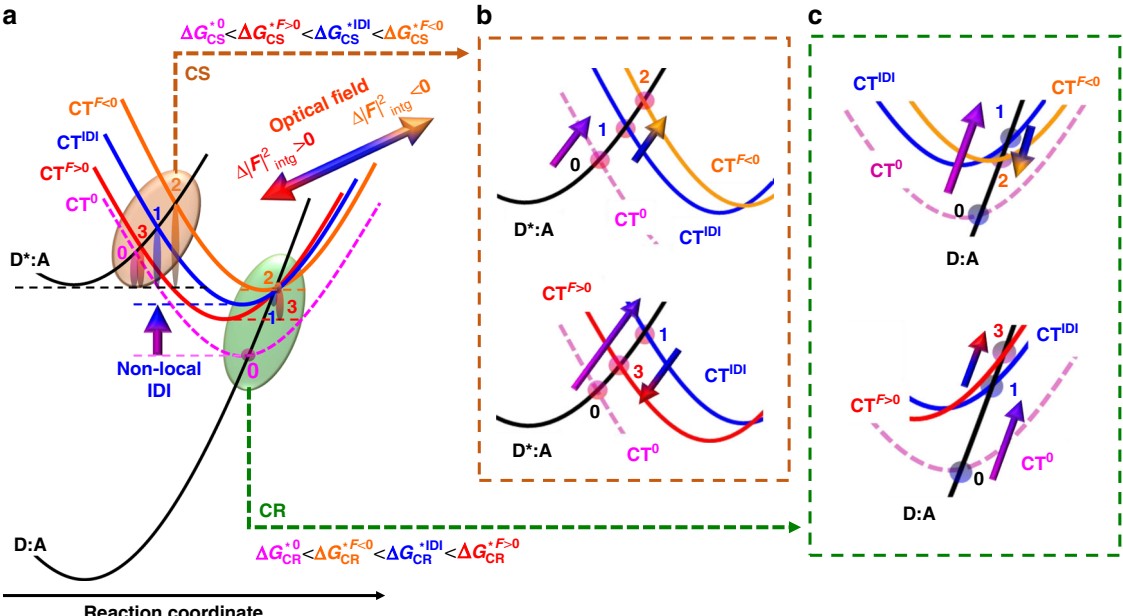

**Fig. 7 Schematic representation of charge-transfer dynamic nanophotonic control by nonlocal enhanced optical-field effects. a** Potential energy profiles as a function of the reaction coordinates. **b, c** close-up views of the brown (CS: charge separation) and green (CR: charge recombination) ellipses, respectively. $S_0$ and $S_1$ are the donor ground and first excited states, while $CT^x$ are the Donor:Acceptor (D:A) system charge-transfer states on fused silica (x: 0), and composite nanostructures without (x: image-dipole interaction—IDI) and with (x: F) the optical-field effect. The vertical stretched ellipses illustrate the activation energy ($\Delta G^*$) variations. Details and magnifications are given in the Supplementary Information. The numbers 0,1,2,3 correspond to those shown in Fig. 6b.

transfer-dynamic-characteristic times are plotted as a function of the optical field intensity. This is consistent with the results presented in Fig. 6b.

Overall, inserting the optical-field effect as a perturbation in Marcus theoretical framework suggests linear relationships between charge-transfer characteristic times and optical-field intensities. These are qualitatively verified and displayed in Fig. 6b. Furthermore, the signs of the charge separation and recombination slopes are consistent with the theoretical developments and discussions having led to Eqs. (8) and (9). This combination of experimental and theoretical results indicates that in composite nanostructures charge-transfer processes can be actively controlled by optical-field intensities.

A control experiment on top of 200-nm-thick Ag films covered by different alumina thicknesses reproduces qualitatively both slopes and amplitude of the charge-transfer dynamic variation with the optical field, as shown in Fig. 6c. We stress that, even when $|F|^2$ is large, simple metal films present a weak image-dipole-interaction contribution because of their single pair structure. Consequently, on top of thick Ag films based composite nanostructures, the slopes are within and barely larger than the experimental and fit errors, for charge separation and recombination, respectively. Nonetheless, the sign of the charge-recombination slope is consistently positive for both 4p and 200-nm-Ag composite nanostructures. Another confirmation that charge-transfer dynamic varies with optical field intensities is shown in Fig. 6d, where control experiments were completed with probe beams of larger power and diameter on top of composite nanostructures covered with 125-nm-thick $Al_2O_3$. Despite a narrower accessible range, the charge-recombination time is again shown to vary with the optical-field intensity.

The alteration of charge-transfer processes due to nonlocal effect based on image-dipole interactions and optical field can be described in terms of shift of energy parabolas as illustrated in Fig. 7. We note that image-dipole interactions move the charge

transfer parabola upwards (pink→blue), and increases the activation energy with the number of pairs in the composite nanostructures for both charge separation and recombination[33]. In contrast, the optical-field effect ($\Delta|F|^2_{intg.}$) can shift the charge-transfer-energy parabola (blue→red) both vertically and horizontally. The former is imposed by the correlation between Eqs. (3) and (4). The latter is consistent with out-of-phase charge-transfer modulations. Noticeably, the nonlocal image-dipole-interaction effect leaves the reorganization energy unaltered, with the $CT^0$ (pink) and $CT^{IDI}$ (blue) parabolas vertically aligned with one another in Fig. 7a. In contrast, the optical-field driven horizontal shift affects both activation and reorganization energies, as it can be better seen in the close-up views presented in Fig. 7b and c. When $\Delta|F|^2_{intg.}<0$, $\tau_{CS}$ increases and $\tau_{CR}$ decreases. The activation energies ($\Delta G^*_{CT}$) follow the same trend, which corresponds to a right shift of the $CT^F$ parabola combined with an upwards translation, as represented by the orange parabola displayed in the top part of Fig. 7b, c. Inversely, when $\Delta|F|^2_{intg.}>0$, $\tau_{CS}$ decreases and $\tau_{CR}$ increases, the resulting $\Delta G^*_{CT}$ variation implies a left-shift of the $CT^F$ parabola compared with the blue $CT^{IDI}$ parabola, as represented by the red parabola displayed in the bottom part of Fig. 7b, c. Magnifications and details are given in the Supplementary Information. For $\Delta|F|^2_{intg.}<0$ and $\Delta|F|^2_{intg.}>0$, we show that the resulting activation energy ($\Delta G^*_{CT}$) variation implies a combined right-upwards and left-downwards shift of the charge-transfer parabola, respectively. The translation-parts along the reaction coordinates impact the reorganization energy. This has not been observed in earlier works, and it results of the simultaneous combination of the optical field and image-dipole-interaction effects.

The generalized description herein proposed is then fully consistent with the experimental charge-transfer behavior and it introduces an electrodynamics component which is usually not

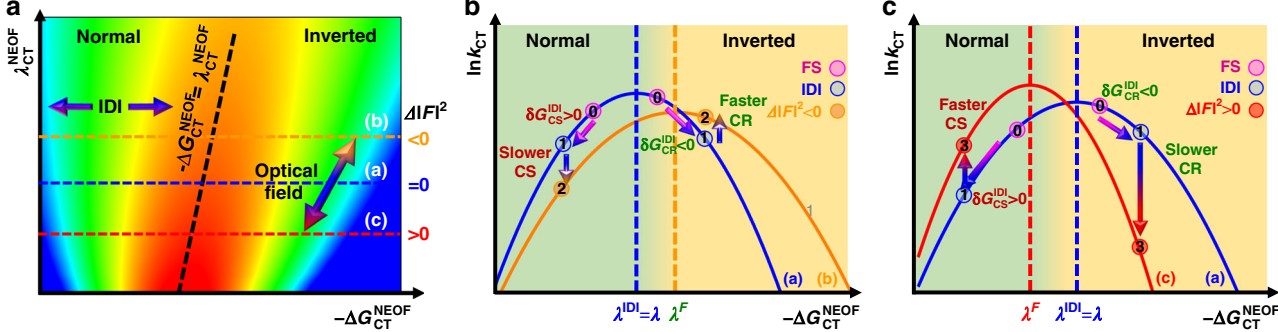

**Fig. 8 Calculated charge-transfer rate on Marcus parabola diagrams. a** False-color plot of the charge-transfer (CT) rate logarithm as a function of the driving force and the reorganization energy. The horizontal and tilted arrows illustrate to the effect of image-dipole interactions (IDI) and of the optical field intensity, respectively. The horizontal and tilted dashed lines represent three iso-optical field cross-sections, and the borderless zone, respectively. Cross-sections of a lead to **b** $\Delta|F|^2 < 0$ and **c** $\Delta|F|^2 > 0$. The vertical dashed lines show the borderless points. The bicolor arrows and circled labels illustrate shifts of the charge-transfer dynamic on fused silica (0) under the nonlocal image-dipole-interaction (IDI) effect (1) and nonlocal enhanced optical field effects (2-3), as described in Figs. 6b and 7.

present in Marcus theory. Noticeably, this formalism bypasses the difficulty of electrical measurements near the composite nanostructures and organic semiconductor material interface, and instead takes advantage of more straightforward calculations relying on the optical constants of each of the materials involved.

Figure 8a presents the calculated charge-transfer rate, $k_{CT}$, on a false-color Marcus parabola diagram as function of the reorganization energy and the driving force based on Eqs. (1) and (4). The position of maximum value of $k_{CT}$, i.e., the barrierless points, moves to the top-right handside as reorganization energy and intrinsic driving force values increase (black dashed line). The image-dipole interaction (IDI) corresponds to a translation parallel to the driving force axis as it does not affect the reorganization energy. The optical field affecting both $\Delta G_{CT}$ and $\lambda_{CT}$ is illustrated by a tilted arrow. Cross-sections of Fig. 8a are presented in Fig. 8b, c, where the horizontal lines (a), (b) and (c) illustrate cases for which $\Delta|F|^2 = 0$, $\Delta|F|^2 > 0$ and $\Delta|F|^2 < 0$, respectively. In the present Donor:Acceptor system, the image-dipole-interaction (IDI) effect pushes both charge separation (CS) and charge recombination (CR) away from the barrierless point as charge separation and recombination are in the normal and inverted regions, corresponding to opposite $\delta G_{CT}^{IDI}$ values. Consequently, both charge separation and recombination slow down under the effect of image-dipole interactions, as illustrated by the labels (0) and (1) on the blue curves. Under the optical-field effect, the charge-transfer rates change from the blue to the orange and red curves, when $\Delta|F|^2<0$ and $\Delta|F|^2<0$, respectively. Both cases represent an out-of-phase charge separation and recombination behavior, which has indeed been observed experimentally.

## Discussion

In summary, the present work demonstrates that composite nanostructures can be engineered for optical fields to both sustain long lasting surface photovoltages and significantly alter Donor: Acceptor charge-transfer dynamics in the solid state. To the best of our knowledge, there are no reports on how surface-photovoltage dynamics can be tuned with composite nanostructures and optical fields have not been explored as charge-transfer-dynamic-alteration triggers. Observing these effects experimentally in a reliable manner is shown to be challenging. Charge-transfer-dynamic modulations were enhanced by taking advantage of nonlocal image-dipole-interaction effects. This was achieved by tuning the structure of the composite nanostructure underneath Donor:Acceptor thin films. A model generalizing

Marcus Theory was developed, including both image-dipole interactions and the optical field intensity, to describe qualitatively the tunability of charge-transfer dynamics with engineered composite nanostructures. Regardless of the region where charge-transfers occur, image-dipole-interactions affect only the driving force, and whether charge-transfer dynamic is accelerated or slowed down depends on the sign of the perturbation $\delta G_{CT}^{IDI}$ as well as the region in which the charge-transfer occurs. The optical field effect is more elaborate as it affects both driving force and reorganization energy. Simplifications can nonetheless be achieved for instance when a Donor:Acceptor system is close to its barrierless point. Then, whether the optical field accelerates or reduces the charge-transfer dynamic is mostly set by the relation between $\left|G_{\lambda-CT}^{IDI}\right|$ and $\sqrt{2\lambda_{CT}k_BT}$, which changes the sign of $\langle\delta_{F2}\rangle$.

The model shows how optical-field perturbations induced simultaneously on driving force and reorganization energy are enhanced by nonlocal image-dipole interactions. The model herein developed provides a qualitatively good agreement with the experimental data displaying out-of-phase variations of charge separation and recombination under nonlocal-enhanced-optical-field (NEOF) effect. It also explains qualitatively well why the NEOF phenomenon could not reliably be observed earlier, for instance on dielectric glass substrates and why the effect is much weaker on single metal layers and when the probe beam intensity and diameter are tuned. Noticeably, the probe beam outweighs the impact of the pump beam and the thickness of the Donor:Acceptor films is also a charge-transfer-dynamic-tuning parameter.

In conclusion, we demonstrated modulations of charge transfer dynamics in the solid state, using organic semiconductor thin films on top of composite nanostructures. These are enhanced and revealed thanks to nonlocal image-dipole-interaction effects associated with metal-dielectric multilayer structures. In the system herein investigated, up to 270% increase in charge separation rate is obtained in the solid state. Charge-separation and recombination modulations are out-of-phase, but the framework herein developed suggests that molecular engineering allows the design of materials in which in-phase charge-transfer-dynamic modulations occur. Indeed, we could rationalize the experimental results within a generalized Marcus theory framework in which image-dipole interactions, as well as optical field, are accounted for. In first approximation, the charge-transfer dynamic on composite nanostructures is shown to be proportional to the incident optical field intensity, which results from nonlocal enhanced optical field (NEOF) effects. We evidenced surface photovoltages, which transients are tuned with the multilayer engineering of the composite nanostructures. They outlast the

pulse width and the repetition rate of standard time-resolved optical spectroscopy, and can participate to the tuning of charge-transfer processes.

The insights gained in this work, including how both driving force and reorganization energy can be affected, open original opportunities to design artificial composite nanostructures both to tune the kinetics of surface photovoltages and to remotely control charge-transfer dynamics in the solid state without having to alter the molecular structure and organization of Donor: Acceptor materials. This is also an exceptional strategy to activate, i.e., to accelerate or to slowdown, charge transfers in the solid state with an external optical trigger. The fundamental knowledge and conceptual understanding revealed by these results will find applications in fields including photonics, optoelectronic devices, material design and chemistry, where selectively altering charge separation and recombination, and more broadly tailoring chemical reaction dynamics, are essential. For instance, this could find exciting applications in optoelectronic devices relying of charge-transfer dynamics, including optically activated nanophotonic switches, photo-modulated chemical reactions in electrochemical cells and electronic-photonic circuitry interfaces.

## Data availability

The data that support the findings of this study are available from the corresponding authors upon reasonable request.

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

## Acknowledgments

This work has been carried out in the framework of the CNRS International Associated Laboratory "Functional nanostructures: morphology, nanoelectronics and ultrafast optics" (LIA NANOFUNC). The authors are thankful to the Institut Parisien de Chimie Moléculaire, UMR 8232, Chimie des Polymères research group for providing the dyad. The authors would like to thank collaborators at the CERC for access to the facilities. K.J.L. and P.A. were supported by fundings of the Ministry of Science, ICT & Future Planning, Korea (201000453, 2015001948, 2014M3A6B3063706). P.A. would like to thank the Canon Foundation in Europe for supporting his Fellowship. K.J.L., S.A.J., and C.G. acknowledge supports from the Bill & Melinda Gates foundation (OPP1119542) and the National Science Foundation (IIP-1701163). L.M.E. acknowledges the Cluster of Excellence 2147 — Complexity and Topology in Quantum Matter (ct.qmat), and E.B. was supported by the Volkswagen Foundation (Grant no. 90 261) and by the Deutsche Forschungsgemeinschaft (DFG, German Research Foundation) within the SFB 1415 — Project-ID 417590517.

## Author contributions

K.J.L. and P.A. conceived and designed the experiments. K.J.L. and S.A.J. prepared the samples under the supervision of C.G. K.J.L. performed both the time resolved optical measurements and their analysis, along with numerical calculations, with the feedback of P.A. K.J.L. and S.J.K. completed and analyzed the spectroscopic ellipsometry measurements, which were discussed with P.A., E.B. and L.M.E. organized the surface photovoltage investigation, with E.B. completing the measurements and analysis, as well as preparing the surface photovoltage related figures. K.J.L. and P.A. developed the theoretical model used to describe the system and to interpret the results. C.G. contributed to optical analyses and discussions. K.J.L. and P.A. discussed on all the results and prepared the figures. P.A. wrote the manuscript and Supplementary Information with K.J.L.'s feedback. All authors commented the manuscript and the reports of the reviewers.

## Competing interests

The authors declare no competing interests.
