## [Peer Review File · Nature Communications]

Reviewers' Comments:

Reviewer #1:

Remarks to the Author:

This is quite an interesting manuscript presenting the enhancement of the charge-transfer modulations via optical near-field of nanostructures. The authors present both theoretical and experimental results, which are in good agreement. The authors carefully analyse the charge separation and recombination times. The authors claim up to 270 % increase in charge separation rate is obtained in the solid-state. The disparity of these times allows for surface photovoltage, which might be used for various applications. The measurement and modelling techniques are appropriate. In general, the manuscript is interesting and the results are exciting. I have just a couple of minor comments: i) the colorbars in Fig.2 are barely seen, ii) can the authors

? iii) My understanding is that the reported results are linked to the so-called Purcell effect. Can the authors elaborate more in relation to this effect? See, for example, "Principles of Nano-Optics" by Lukas Novotny and Bert Hecht.

Reviewer #2:

Remarks to the Author:

Lee et al produce a rather confusing manuscript, where it is very difficult to understand the physics beyond the venerable Marcus theory. The authors use many acronyms for well understood concepts, which then obscures their meaning. Moreover, they use language such as "the electron field intensity" making it unclear whether they mean the field amplitude (E) or light intensity (E^2). They analyze their result with the Marcus theory, but it is not at all clear what are the appropriate electron potentials, and how they are obtained. The authors can construct potentials that explain their observations, but it is difficult to test the validity, and even if valid application of the Marcus theory seems hardly to have a broad appeal. Perhaps it would have been interesting 40 years ago, but resurrecting these well-worn concepts seems not very appealing. To me it seems like the authors are trying too hard to publish in a well cited journal by giving well-worn concepts new and obscure names, making the entire manuscript a chore to read.

Reviewer #3:

Remarks to the Author:

Review of the manuscript "Optical-Field Driven Charge-Transfer Modulations near Composite Nanostructures" by Lee et al.

The study by Lee et al. demonstrates that one can engineer composite nanostructures supporting rationally designed nonlocal dielectric environments, which optically trigger photoinduced modulation of the charge transfer dynamic in the solid state. The reported lamellar metal/dielectric nanostructures enable an intriguing out-of-phase behavior of charge separation and charge recombination dynamics, along with control of the charge transfer dynamics with varying optical-field amplitude. Thus, the 270% increase in the charge separation rate has been obtained in organic semiconductor thin films. Furthermore, it has been shown that the fabricated metal-dielectric nanostructures allow for surface photovoltages to be created and outlast the optical pulse illumination. The modified Marcus theory, taking into account non-local dielectric environments (in particular, mirror images of dipoles), has been developed to explain the modulations in the charge transfer dynamics. The authors expect the discovered phenomenon to find applications in photochemistry and photoelectronics.

The study is very interesting both in terms of fundamental science and potential applications. The developed theory is in a good agreement with the experiment. The importance of this work warrants its publication in Nature Communications. However, first, the manuscript should be improved as suggested below.

Overall, the manuscript is difficult to read. This is partly because the main body of the manuscript (6 pages) is not self-contained and the reader is continuously referred to the 20-pages-long Supplemental Information section. This hinders smooth reading of the paper.

Another problem, of a lesser scale, is an excessive usage of acronyms. These acronyms make reading difficult.

Lee and Andre are the co-authors of the related recent paper [33]. The authors should clearly explain what in the manuscript is new and what is similar to Ref. [33].

The kinetics of Figs 1b and 1c are essential to this work. The authors should explain in the main text how these kinetics were taken and how the measurement results were interpreted.

“Overall, maximum CS and CR variations correspond to factors of 3.7 and 1.7, respectively, when compared to FS data. These factors reach ~1.57 and ~1.44 when comparing CTDs obtained on the CNSs, corresponding to relative variations of 57% and 44%, respectively.”
– this is unclear. What is compared to what?

Fig. 2 : The conclusions drawn from these calculations should be emphasized more clearly.

“Evidence of Long-Lasting Surface Photovoltages. For any optical effect to play a role on CTDs, it should outlast both the pulse width and the repetition rate time scale.”
- It should be explained why the “outlasting” is so important and what is the physical reason for this “outlasting”.

“In principle, interfaces, which exhibit an optically accessible depletion zone due to an energy band bending can show a change in the electrical potential upon illumination due to charge exchange processes between the bands or between trap states and the bands,” – the schematics of the band diagram referenced in this sentence is needed.

The reaction’s energy diagram showing the characteristic energies described by Eqs. 1 and 2 is needed. This will greatly benefit the readers, who are not familiar with the Marcus theory. Can the energy terms described by Eqs. 3 and 4 be shown in the same diagram without making the figure too busy?

The conclusions based on Eqs. 5-7 and Fig. 4 are important. They should be explained more clearly.

Shifts of parabolas in Fig. 5 are difficult to follow. I suggest the authors to clearly explain the conclusions drawn from these shifts and move the details of the derivations to an Appendix. This recommendation can be extended to the whole Marcus theory section of the paper.

Answer to the reviewer’s comments – August 2020

Reviewer #1 (Remarks to the Author):

This is quite an interesting manuscript presenting the enhancement of the charge-transfer modulations via optical near-field of nanostructures. The authors present both theoretical and experimental results, which are in good agreement. The authors carefully analyse the charge separation and recombination times. The authors claim up to 270 % increase in charge separation rate is obtained in the solid-state. The disparity of these times allows for surface photovoltage, which might be used for various applications. The measurement and modelling techniques are appropriate. In general, the manuscript is interesting and the results are exciting. I have just a couple of minor comments:

We would like to thank Reviewer #1 for the thorough reading and comments of both the main text and the SI. We are pleased to read that Reviewer #1 recognizes that our manuscript is interesting, the results are exciting, and that the theoretical and experimental results are in good agreement.

1. The colorbars in Fig.2 are barely seen,
We have modified the Figure 2 (now Figure 3) to make the color bars more readable. The font size was increased, we inserted a white background for the letters and surrounding for the colorbars.

Figure 3 | Optical modulation induced by the substrate structure. **a,b,c** False-color plot of the optical field intensity, $|F(z)|^2$, as a function of incident wavelengths and position within the sample. **d,e,f** Optical-field intensity distribution within the sample corresponding to a cross section at 725 nm of **(a,b,c)** with the background colors representing the sample composition. Fused silica substrate **(a,d)**, 4p-composite nanostructure (CNS) with an Al_2O_3 top layer of 10 nm **(b,e)**, and 100 nm **(c,f)**, respectively, and Donor:Acceptor (D:A) thin film. The horizontal lines materialize the different layers of materials.

2. Can the authors [redacted]?

Wondering whether or not we can optimize the nanophotonic structures at any operational wavelength is an excellent question. From a methodological point of view, the short answer is “yes”. However, this is a complex and very exciting point, on which we are currently working [redacted]

Indeed, [redacted]

[redacted], and this goes beyond the scope of the present [redacted]

[redacted] will require a separate future study from our current report of out-of-phase charge transfer dynamics and the fundamental mechanism investigation, which we have submitted to Nature Communications.

3. My understanding is that the reported results are linked to the so-called Purcell effect. Can the authors

elaborate more in relation to this effect? See, for example, "Principles of Nano-Optics" by Lukas Novotny and Bert Hecht.

We have studied the oscillating behavior of Purcell factor in a previous work where we used a set of photoluminescent chromophores in PMMA matrices and we identified the regime in which Purcell factor shows a strong dependence on the number of metal-dielectric pair (ref. 29: Lee, K. J., Lee, Y. U., Kim, S. J., André, P., *Hyperbolic Dispersion Dominant Regime Identified through Spontaneous Emission Variations near Metamaterial Interfaces*. Adv. Mater. Interfaces 5, 1701629 (2018)). However, we would like to emphasize that the charge transfer phenomenon, which we report in this Nature Communications study, is not directly related to Purcell effect since the present organic semiconductors don't emit photons. Therefore, we could not use the Purcell formalism presented in the reference mentioned by Reviewer #1, and instead we developed a model generalizing Marcus theory that governs overall charge transfer processes.

Nonetheless, Reviewer #1 raises a clearly important point, which we tried to further take into account by adding the following sentence in the manuscript p. 2 col. 2 "*At least because the present organic-semiconductor-charge transfers are non-emissive, we can also already conclude that the herein reported phenomenon is unrelated to Purcell effects⁵⁴, which we have previously studied on composite nanostructures²⁹.*".

In summary, we are grateful for the very constructive comments of Reviewer #1, we believe that we have fully addressed them and that the paper is suitable for publication in Nature Communications at this point.

Reviewer #2 (Remarks to the Author):

Lee et al produce a rather confusing manuscript, where it is very difficult to understand the physics beyond the venerable Marcus theory.

We are sorry that Reviewer #2 felt that the manuscript was not presented clearly confusing. We have gone through all the comments very carefully and our reply is as below.

1. The authors use many acronyms for well understood concepts, which then obscures their meaning.

We acknowledge that both Reviewer #2 and #3 felt that the 14 acronyms used in the previous version of the manuscript did not help its reading. We have checked all acronyms used in the paper and have listed them below,

	Previous version	Current Version used alone
(1) Donor:Acceptor (D:A)	20x	still used in the figures 0x
(2) charge transfer dynamic (CTD)	48x	0x
(3) composite nanostructure (CNS)	50x	still used in the figures 0x
(4) charge separation (CS)	25x	still used in the equations and figures 0x
(5) charge recombination (CR)	37x	still used in the equations and figures 0x
(6) charge transfer (CT)	33x	still used in the equations 5x
(7) surface photovoltage (SPV)	25x	still used in the figures 0x
(8) image-dipole interactions (IDI)	35x	still used in the equations and figures 1x
(9) nonlocal enhanced optical field (NEOF)	5x	still used in the equations and figures 1x
(10) triphenylene (TriPh)	4x	now removed 0x
(11) perylene diimide (PerDi)	3x	now removed 0x
(12) fused silica (FS)	12x	now removed 0x
(13) transient absorption signals (TAS)	1x	now removed 0x
(14) transient absorption (TA)	1x	now removed 0x

To address these convergent opinions, we have eliminated 11 of these acronyms (D:A, CTD, CNS, CS, CR, SPV, TriPh, PerDi, FS, TAS, and TA) from the text. In addition, each time acronyms were used we made sure that its meaning was reminded in the vicinity, we added a list of acronyms in the appendix, which complements the one already in the SI; finally, we also refrained from using acronyms alone in captions.

We hope this approach both addresses the reviewers' concerns, makes the manuscript more readable.

2. Moreover, they use language such as "the electron field intensity" making it unclear whether they mean the field amplitude (E) or light intensity (E^2).

We would also be very much confused by the above language and this is certainly not the way we would use it. We thought that it could have happened that a typo led the manuscript to include the expression "*the electron field*

intensity". However, we could not find it anywhere, neither in the manuscript, nor in the SI files, which we had previously submitted. We do not rule out a typo on our behalf, but we would be grateful if Reviewer #2 could let us know where he/she has read this expression in our manuscript and/or its SI.

The background of the co-authors being complementary, we believe that the wording we chose to use is rather neutral and consensual including "*electromagnetic wave*", "*electromagnetic field*" and "*optical field intensity*" in both singular and plural forms. We note that we had already written $|F|^2$ directly in the previously submitted documents, and naturally $|F|^2$ explicitly refers to the optical field intensity.

Confused by the comment of Reviewer #2, but hoping to increase the clarity of our manuscript, we have now included in the text p. 3 col. 1 that optical-field intensity is also referred to as light intensity. In addition to further clarify the terminology used in this manuscript, we replaced "*optical field amplitude*" with "*optical field intensity*" in the abstract since this work is only dealing with the influence of field intensity.

We hope these efforts address Reviewer #2 in a satisfactory manner.

3. They analyze their result with the Marcus theory, but it is not at all clear what are the appropriate electron potentials, and how they are obtained.

Indeed, we use Marcus theory to analyze our result. However, Marcus theory and electromagnetic theory have each been used on its own and separately, and then part of our work aims first at linking them together. Once the formalism established, we have shown that the relatively large set of experimental results and the theoretical framework, we have both obtained, are in good qualitative agreement. We stress that we also work with a relatively large set of experimental results, and not a single measurement. The combination of both elements means that we do not need to use electron potential values, which would be questioned. We understand the concern of Reviewer #2, had we used electron potential values, however, one of the advantages of the methodology we have implemented is that it does not rely on guessing appropriate or convenient electron potential values, which then cannot be questioned.

To build on Reviewer #2's comment, we would like to add that we believe that it should be possible to use the framework herein developed to eventually obtain accurate and optical field independent electron potential values. However, this goes far beyond the scope of the present manuscript, and it will be the subject of further long-term investigations.

4. The authors can construct potentials that explain their observations, but it is difficult to test the validity, and even if valid application of the Marcus theory seems hardly to have a broad appeal. Perhaps it would have been interesting 40 years ago, but resurrecting these well worn concepts seems not very appealing. To me it seems like the authors are trying too hard to publish in a well cited journal by giving well-worn concepts new and obscure names, making the entire manuscript a chore to read.

We would like to emphasize that even recently, several research papers based on Marcus theory have been published in high impact journals including from the Nature research journals.¹⁻¹¹ Rudolph A. Marcus himself is still participating to such research activity based on his theory.¹² Overall, valid applications of the Marcus theory definitely have a broad appeal.

Besides, we would like to stress that we are not just using Marcus theory, but we are developing a more generalized formalism to include in Marcus theory electromagnetism theory parameters such as optical field intensity, which to the best of our knowledge has not been done before the present manuscript.

Still, we would welcome any literature references, which Reviewer #2 could provide us with, showing that our work has already been completed years ago. We could not find them but we would certainly consider them carefully, if they were made available.

On the other hand, we also took note that Reviewer #2 found the manuscript confusing and hard to understand beyond the physics of Marcus theory. Then, we hope that the alterations we made to the manuscript will help Reviewer #2 to see the novelty and interest of our work, for instance

- i-* when linking Marcus theory and electromagnetism theory,
- ii-* the original influence of optical fields on charge dynamics in solid films,
- iii-* the opportunities offered by adjusting nanostructures to control charge dynamics in solid state optoelectronic devices in which Marcus theory is quite relevant, and we believe it is both reinforced and extended by the present work.

We also hope that Reviewer #2 will be convinced that optoelectronic devices are part of the broad appeal that the present results can bring to the scientific community.

Overall, we hope that the changes we have implemented to the manuscript make this work more accessible to the wide readership of Nature Communications.

Reviewer #3 (Remarks to the Author):

The study by Lee et al. demonstrates that one can engineer composite nanostructures supporting rationally designed nonlocal dielectric environments, which optically trigger photoinduced modulation of the charge transfer dynamic in the solid state. The reported lamellar metal/dielectric nanostructures enable an intriguing out-of-phase behavior of charge separation and charge recombination dynamics, along with control of the charge transfer dynamics with varying optical-field amplitude. Thus, the 270% increase in the charge separation rate has been obtained in organic semiconductor thin films. Furthermore, it has been shown that the fabricated metal-dielectric nanostructures allow for surface photovoltages to be created and outlast the optical pulse illumination. The modified Marcus theory, taking into account non-local dielectric environments (in particular, mirror images of dipoles), has been developed to explain the modulations in the charge transfer dynamics. The authors expect the discovered phenomenon to find applications in photochemistry and photoelectronics.

The study is very interesting both in terms of fundamental science and potential applications. The developed theory is in a good agreement with the experiment. The importance of this work warrants its publication in Nature Communications.

We thank Reviewer #3 for the thorough reading of our manuscript and the constructive comments shared with us. We have carefully considered them to both answer them and to adjust the manuscript accordingly.

However, first, the manuscript should be improved as suggested below.

1. Overall, the manuscript is difficult to read. This is partly because the main body of the manuscript (6 pages) is not self-contained and the reader is continuously referred to the 20-pages-long Supplemental Information section. This hinders smooth reading of the paper.

It is true that the previous version of the manuscript referred 18 times to the SI, mostly to figures and a table to support arguments and offer further elements to the readers. It is often difficult to find a balance between preparing a short manuscript and providing further information and support material in the SI, especially when we aim at a broad readership. We appreciate the suggestion of the Reviewer #3 that, on this occasion, the balance should be altered. As a consequence, rather than distributing the calls to the SI throughout the text which can be practical but can also be a distraction from the report content and give the impression that the manuscript is not self-contained,

- we reduced these calls and better described the SI content in the Additional Information section,
- we removed references to SI-V and references to SI-Fig. 4, SI-Fig. 5, SI-Fig. 7, SI-Fig. 8, SI-Fig. 9, SI-Fig. 14, SI-Fig. 18 and SI-Fig. 19.
- we moved from the SI to the main manuscript
 - o SI-Table 1, which is now displayed as Table 1,
 - o SI-Fig. 4 and SI-Fig. 14 have been merged into a new Figure 2 (see on the right hand side),
 - o SI-Fig. 18 and SI-Fig. 19 have been merged with Fig. 5 (formerly Figure 4 ; see next page).

Figure 2 | Analysis of charge-transfer-dynamic and optical field in composite nanostructure. **a**, Variations of charge separation (CS) and charge recombination (CR) ratio as a function of the Al_2O_3 top layer thickness. **b**, Optical field intensity distributions of both pump (325 nm) and probe (725 nm) beams across the Al_2O_3 top-covers.

Figure 6 | Analysis of charge-transfer-dynamic-nanophotonic control by nonlocal enhanced optical field effects. **a**, Variations of charge separation (CS, ●) and charge recombination (CR, ●) normalised dynamics and integral of the optical field (●) as a function of the Al_2O_3 top layer thickness. **b**, Linear charge separation (CS: ●) and recombination (CR: ●) dynamics dependence with the integral of optical field intensity of a 725 nm incident beam across the organic Donor:Acceptor thin films on top of 4p composite nanostructures. The bottom pink data points were obtained on fused silica substrate, the vertical and horizontal arrows materialize nonlocal image-dipole-interaction (IDI) effect and nonlocal enhanced optical field effects, labelled (0), (1) and (2-3), respectively. **c**, Linear charge separation (CS: ●) and recombination (CR: ●) dynamics dependence with the integral of 725 nm optical field intensity across the organic Donor:Acceptor thin films on top of 200 nm Ag composite nanostructures. **d**, Charge recombination dynamics plotted against the integral of 725 nm optical field intensity for different probe beam powers (CR-P: ●) and beam diameters (CR-D: ●) measured on top of Ag- Al_2O_3 composite nanostructures covered with 125 nm thick Al_2O_3 . The dashed lines are guide for the eyes.

Then, to address this comment, four figures and one diagram were inserted in the main manuscript and the text is now a bit longer.

We hope that Reviewer #3 will find these changes appropriate.

2. Another problem, of a lesser scale, is an excessive usage of acronyms. These acronyms make reading difficult. We acknowledge that both Reviewer #2 and #3 felt that the 14 acronyms used in the previous version of the manuscript did not help its reading. We have checked all acronyms used in the paper and have listed them below,

	Previous version	Current Version
(1) Donor:Acceptor (D:A)	20x	still used in the figures 0x
(2) charge transfer dynamic (CTD)	48x	still used in the figures 0x
(3) composite nanostructure (CNS)	50x	still used in the figures 0x
(4) charge separation (CS)	25x	still used in the equations and figures 0x
(5) charge recombination (CR)	37x	still used in the equations and figures 0x
(6) charge transfer (CT)	33x	still used in the equations 5x
(7) surface photovoltage (SPV)	25x	still used in the figures 0x
(8) image-dipole interactions (IDI)	35x	still used in the equations and figures 1x
(9) nonlocal enhanced optical field (NEOF)	5x	still used in the equations and figures 1x
(10) triphenylene (TriPh)	4x	now removed 0x
(11) perylene diimide (PerDi)	3x	now removed 0x
(12) fused silica (FS)	12x	now removed 0x
(13) transient absorption signals (TAS)	1x	now removed 0x
(14) transient absorption (TA)	1x	now removed 0x

To address these convergent opinions, we have eliminated 11 of these acronyms (D:A, CTD, CNS, CS, CR, SPV, TriPh, PerDi, FS, TAS, and TA) from the text. In addition, each time acronyms were used we made sure that its meaning was reminded in the vicinity, we added a list of acronyms in the appendix, which complements the one already in the SI; finally we also refrained from using acronyms alone in captions.

We hope this approach both addresses the reviewers' concerns, makes the manuscript more readable, and we would like to thank them for this comment.

3. Lee and Andre are the co-authors of the related recent paper [33]. The authors should clearly explain what in

the manuscript is new and what is similar to Ref. [33].

We thank the Reviewer for this suggestion. In p. 1 col. 2, we have substantially altered the introduction to the results section, which now reads as:

“In a previous and related work³³, we focused on the number of pairs of composite nanostructures and inserted image-dipole interactions in Marcus theory framework to show that both charge separation and recombination slowed down with the number of metal-dielectric pairs. Albeit exciting in itself, the limitations of this earlier work included, however, to consider the slowdown of charge-transfer-state dynamics as only controlled by one substrate-structural parameter; it could not identify ways to impact differently separation and recombination characteristic times and only the driving force seemed affected. In the present work, we address these limitations by focusing on 4-pair composite nanostructures, and we use the thickness of the top Al₂O₃ spacer to reveal unexpected optical field effects. We show that engineered composite nanostructures provide unprecedented optical control over charge transfer dynamics in the solid-state. On such nanostructures, we evidence that charge separation and recombination dynamics display out-of-phase modulations, one increasing when the other decreases. This original, and so-far unreported behaviour is shown to result from a long lasting optical-field effect mediated by surface photovoltages, an overlooked parameter in multilayered nanostructures, herein enhanced by nonlocal image-dipole interactions (IDI). Out-of-phase charge-transfer-dynamic modulations are successfully described after introducing optical-field intensity in a generalised Marcus theory framework. This original theoretical contribution reproduces the trend of the experimental data and it also shows that both driving force and reorganization energy are affected. The present results shine new lights on the potentials offered by nonlocal environments and by composite nanostructures engineering showing how charge-transfer dynamics, essentially governed by thermodynamic mechanisms, can be controlled in the solid-state by electromagnetic fields acting as remote actuators with differentiated impacts on charge separation and charge recombination.”

4. The kinetics of Figs 1b and 1c are essential to this work. The authors should explain in the main text how these kinetics were taken and how the measurement results were interpreted.

We agree that the description of Figs 1b and 1c are essential to this work and this section was expanded from 8 lines (4 sentences) to 21 rows (10 sentences). It now reads as:

“The sub-picosecond transient absorption spectra presented in Figures 1b and 1c are generated by pumping at 325 nm and monitoring at 725 nm and constant time intervals. The relative transmittance variation ($\Delta T/T$) curves are obtained from annealed dyad thin films on composite nanostructures for short and long delay times, respectively. They display typical transient absorption features resulting from the formation and disappearance of photo-induced charge-transfer states. In Figure 1b, the $\Delta T/T$ signal varies progressively as acceptor radical anions are formed, corresponding to photoinduced charge separation (CS) and leading to the formation of charge-transfer states. These are equivalently described as dipole moments between triphenylene and perylene diimide charged units. A plateau is reached within 3 ps, when no more charge-transfer states induced by the pump pulse are formed. Over a much longer time scale and for longer time interval, the signal recovery displayed in Figure 1c monitors charge recombination (CR). It corresponds to the progressive disappearance of charge-transfer states, until a plateau is reached within 700 ps. The insets show that both dynamics are properly described with single exponential characteristic times. The kinetics obtained on a range of composite nanostructures are presented in Table 1.”

5. “Overall, maximum CS and CR variations correspond to factors of 3.7 and 1.7, respectively, when compared to FS data. These factors reach ~1.57 and ~1.44 when comparing CTDs obtained on the CNSs, corresponding to relative variations of 57% and 44%, respectively.”

– this is unclear. What is compared to what?

To facilitate its reading, we adjusted this sentence in the manuscript to:

“Overall, when compared to fused silica data, charge separation and recombination on top of composite nanostructures varied by a maximum factor of 3.7 and 1.7, respectively. They correspond to spacer thicknesses of 125 nm and 10 nm, with the ratios of 0.85 ps to 0.23 ps and 391 ps to 226 ps, as shown in Table 1. When comparisons are completed between kinetics obtained on composite nanostructures, these factors reach ~1.57 and ~1.44, corresponding to relative variations of 57% and 44%, respectively. This charge separation maximum variation is obtained between spacer thicknesses of 125 nm (0.85 ps) and 10 nm (0.55 ps), while this charge recombination maximum is observed between spacer thicknesses of 10 nm (391 ps) and for 300 nm (269 ps).”

6. Fig. 2: The conclusions drawn from these calculations should be emphasized more clearly.

We thank Reviewer #3 to advise emphasizing the conclusions drawn from these calculations associated with Fig. 2 (now Fig.3). We have now modified the transition from this section to the next with:

“These calculations suggest that optical field intensities within the nanostructures and organic semiconductor thin film can be tuned by changing the thickness of the top Al₂O₃ layer. In the present situation, where there is no

emission from the organic semiconductor molecules, the calculated variations of the optical fields could be linked directly or indirectly with the oscillatory phenomena reported in Figure 1. They could also become an attractive tool to qualitatively model the charge-transfer-dynamic modulations herein reported, even though at the likely condition of identifying a bridge between intensity and kinetics. This is what the following sections explore.”

7. “Evidence of Long-Lasting Surface Photovoltages. For any optical effect to play a role on CTDs, it should outlast both the pulse width and the repetition rate time scale.”

- It should be explained why the “outlasting” is so important and what is the physical reason for this “outlasting”. During the transient-absorption experiments completed with a pump-probe technique, a light field is only present when the (short) pump and probe pulses are incident on the sample. In the time delay between the pulses, where most of the charge transfer dynamics must occur, there should not be any light field. In order to address this issue, we demonstrated that surface photovoltage induced by probe pulse outlasts both the pulse duration and even the repetition rate time scale. This indicates that an optical field formed by a pulse laser beam can constantly affect the charge transfer dynamics via inducing a surface photovoltage. To clarify this, we have added sentences in the surface photovoltage section as follows,

“In pump-probe transient experiments, the light field is only present when the pump and probe short pulses are passing through the sample. During the time delay between the pulses, i.e. when the charge transfers occur, there should not be any light. Consequently, for any optical effect to play a role on composite nanostructure’s charge-transfer dynamics, it should outlast the 60 fs pulse width and the 5 kHz repetition-rate-time scale of the incident-transient-absorption-optical-beam parameters, as well as the few hundred picoseconds associated with the charge-recombination-time scale. As discussed above, an incident optical field can provide the oscillatory behaviour, but the dynamics of the optical beam used in the transient-absorption experiments are so fast that it then needs to trigger a build-up, which will impact charge-transfer dynamics long after the incident waves have vanished from the nanostructures. The build-up phenomenon ought to work for a much longer time scale than those mentioned above. Combining the charge-transfer-dynamic oscillations with the requirement to outlast the incident waves and charge-transfer-characteristic times, points toward an indirect optical effect mediated by the architecture of the composite nanostructures - a mediation which has not been considered before. In this context, surface photovoltages are exceptionally relevant as they are photoinduced and present – at least in a number of wider-gap materials – much longer characteristic times than those of the incident waves. The validity of this hypothesis was demonstrated by completing Kelvin-probe time-resolved surface photovoltage measurements on three bare exemplary composite nanostructures.”

8. “In principle, interfaces, which exhibit an optically accessible depletion zone due to an energy band bending can show a change in the electrical potential upon illumination due to charge exchange processes between the bands or between trap states and the bands,” – the schematics of the band diagram referenced in this sentence is needed.

Following up on this suggestion, we have added a schematic illustration of a tentative band diagram of the present system as SI-Figure 3 |:

SI-Figure 3a shows a possible configuration of the Al_2O_3 -air interface with a downward band bending towards the surface corresponding to a hole-depleted interface in the p-type-like oxide.¹³ Under illumination, the incident photons excite electrons from the valence band to the trap levels, the corresponding holes move towards the bulk due to the internal field, and the charge redistribution between surface and bulk alters the band bending and the surface potential. The difference of the total surface band bendings in the dark and under illumination corresponds to the surface photovoltage (SPV).

SI-Figure 3b shows a possible configuration of the composite nanostructure-air interface with the Al_2O_3 configuration influenced by the Ag layers underneath, which can also build up a depletion zone due to ionic bonding from the silver layer to outer oxygen ions, which means an electron transfer from Ag to O, the latter ions act thus as acceptors.¹⁴⁻¹⁶ In this case, all optically accessible buried Ag/ Al_2O_3 interfaces can also show an SPV contribution (for the sake of clarity, the SPV process is only sketched for the surface); the measured SPV is a cumulative effect, since all interfaces of the structure are connected in series.

Figure SI-3 | Schematic illustration of a tentative band diagram of the present system. a, aluminium oxide, b, composite nanostructure. With E_{vac} the vacuum level, E_c the conduction level, E_F the Fermi level, E_t a trap level distribution, E_v the valence level. The red dashed curve illustrates an SPV effect on band bending, the light red cone illustrates the SPV range under the influence of the optical field effect. The areas in grey, grey to white gradient and yellow stand for the oxide, hole-depletion region, and metal, respectively.

9. The reaction's energy diagram showing the characteristic energies described by Eqs. 1 and 2 is needed. This will greatly benefit the readers, who are not familiar with the Marcus theory. Can the energy terms described by Eqs. 3 and 4 be shown in the same diagram without making the figure too busy?

We thank the reviewer for suggesting to add a basic schematic description of Marcus theory based on Eqs. (1) and (2) to assist readers unfamiliar with Marcus theory. Even though it has not been possible to insert the energy terms described by Eqs. (3) and (4) in the same diagram without making it too busy, the energy parabola diagram is now inserted as Figure 5 on p. 4 col. 2, associated with the following sentence "For illustration purposes, these are represented in Figure 5, with the parabolic energy levels of the reactant and the product shifted along the reaction coordinates and energy axis."

Figure 5 | Schematic representation of charge transfer. Energy parabola shift along the energy axis and reaction coordinates showing the Gibbs free energy gain (ΔG_{CT} , driving force), the activation energy (ΔG_{CT}^*), and the reorganization energy (λ_{CT}).

A slightly more developed version has also been inserted in SI-V.1. and includes the following section:

"The driving force, ΔG , is basically quantified by the offset between the minimum energy of reactant and that of the product. In the present case, these are the Donor's excited state ($D^*:A$) and the charge transfer state ($D^+:A^-$), while ΔG is given by the offset between the lowest unoccupied molecular orbital (LUMO) of the Donor and the LUMO of the acceptor. λ is the reorganization energy, which is an amount of energy needed to reorganize the nuclear configuration when a charge transfer takes place. This is indicated as the energy difference between the product's minimum energy and the reactant energy at nuclear position of the product ground state."

10. The conclusions based on Eqs. 5-7 and Fig. 4 are important. They should be explained more clearly.

In order to explain more clearly, we have added sentences as follows,

"Overall, inserting the optical-field effect as perturbation in Marcus-theoretical framework suggests linear relationships between charge-transfer characteristic times and optical-field intensities. These are qualitatively

verified and displayed in Figure 6b. Furthermore, the signs of the charge separation and recombination slopes are consistent with the theoretical developments and discussions having led to Eq. (8) and (9). This combination of experimental and theoretical results indicate that in composite nanostructures the charge transfer processes can be actively controlled by optical field intensities.”

11. Shifts of parabolas in Fig. 5 are difficult to follow. I suggest the authors to clearly explain the conclusions drawn from these shifts and move the details of the derivations to an Appendix. This recommendation can be extended to the whole Marcus theory section of the paper.

This is an interesting and challenging recommendation, which effects we have carefully considered. The development of Marcus theoretical framework is an important part of the work and moving it to an Appendix would hide this contribution while also implying that theoretical developments are somehow of a lesser importance than experimental results. We feel, however, that between experimental and theoretical, any associated value difference would be subjective, and we are keen not to engage in this direction. Furthermore, and as often when mixing experimental and theoretical results, we understand that it can be difficult to strike a balance between talking to both communities and neither repelling nor ignoring one of them.

The readership of Nature Communications is composed of members from both communities. Consequently, to address this balancing act, we have kept the harder mathematical presentations for the SI, while introducing the key-theoretical elements in the main manuscript. By doing so, the less-mathematical-physics-oriented readers can skip the heaviest part while still get glimpse of theoretical considerations, and the more-mathematical-physics-oriented readers can be attracted into digging in the SI.

The fact that Reviewer #1 has highlighted the presentation of “both theoretical and experimental results”, a careful analysis, and that “the measurement and modelling techniques are appropriate” could be seen as supportive of the balance we reached and we then feel that appendix should not host the whole Marcus theory section of the paper. Nonetheless, we have adjusted the parabola shifts in Figure 7 (previously Figure 5), which can now be read in light of the reaction’s energy diagram showing the characteristic energies described by Eqs. (1) and (2), which was inserted as Figure 5 thanks to the comment 9 of Reviewer #3. We hope these alterations will contribute at addressing the comment 11 of Reviewer #3.

Overall, we are grateful for the very constructive comments of Reviewer #3, we hope to have fully addressed them, and that he/she will support the publication of the present work in Nature Communications.

References

1. Yamashita, Y., *et al.*, Efficient molecular doping of polymeric semiconductors driven by anion exchange. *Nature* **572**, 634-638 (2019).
2. Yuan, L., *et al.*, Transition from direct to inverted charge transport Marcus regions in molecular junctions via molecular orbital gating. *Nat. Nanotechnol.* **13**, 322-329 (2018).
3. Pace, N. A., *et al.*, Slow charge transfer from pentacene triplet states at the Marcus optimum. *Nat. Chem.* **12**, 63-70 (2020).
4. Righetto, M., *et al.*, Hot carriers perspective on the nature of traps in perovskites. *Nat. Commun.* **11**, 2712 (2020).
5. Atxabal, A., *et al.*, Tuning the charge flow between Marcus regimes in an organic thin-film device. *Nat. Commun.* **10**, 2089 (2019).
6. Lu, Y., Kundu, M., Zhong, D., Effects of nonequilibrium fluctuations on ultrafast short-range electron transfer dynamics. *Nat. Commun.* **11**, 2822 (2020).
7. Zhong, Y., *et al.*, Sub-picosecond charge-transfer at near-zero driving force in polymer:non-fullerene acceptor blends and bilayers. *Nat. Commun.* **11**, 833 (2020).
8. Taylor, N. B., Kassal, I., Generalised Marcus theory for multi-molecular delocalised charge transfer. *Chem. Sci.* **9**, 2942-2951 (2018).
9. Sowa, J. K., Mol, J. A., Gauger, E. M., Marcus Theory of Thermoelectricity in Molecular Junctions. *J. Phys. Chem. C* **123**, 4103-4108 (2019).
10. Roy, S., Baer, M. D., Mundy, C. J., Schenter, G. K., Marcus Theory of Ion-Pairing. *J. Chem. Theory Comput.* **13**, 3470-3477 (2017).
11. Volchkov, V. V., *et al.*, Intramolecular photo-driven electron transfer in the series of DMABN related compounds with para-substituted acceptors. Study of the rate constants by Marcus theory. *J. Phys. Org. Chem.* **33**, (2019).
12. Zhao, D., *et al.*, Monitoring Electron-Phonon Interactions in Lead Halide Perovskites Using Time-Resolved THz Spectroscopy. *ACS Nano* **13**, 8826-8835 (2019).

13. Robertson, J., Falabretti, B., Band offsets of high K gate oxides on III-V semiconductors. *J. Appl. Phys.* **100**, 014111 (2006).
14. Eremeev, S. V., Schmauder, S., Hocker, S., Kulkova, S. E., Investigation of the electronic structure of Me/Al₂O₃(0001) interfaces. *Physica B* **404**, 2065-2071 (2009).
15. Zhukovskii, Y. F., Kotomin, E. A., Herschend, B., Hermansson, K., Jacobs, P. W. M., The adhesion properties of the Ag/ α -Al₂O₃ interface: an ab initio study. *Surf. Science* **513**, 343-358 (2002).
16. Deng, H., *et al.*, Nature of Ag Species on Ag/ γ -Al₂O₃: A Combined Experimental and Theoretical Study. *ACS Catal.* **4**, 2776–2784 (2014).

Reviewers' Comments:

Reviewer #1:

Remarks to the Author:

I'm quite satisfied with the authors' response. The revised manuscript and supplemental materials are more insightful now. I do share some of the concerns of the Reviewer #2, but find that the authors did a great job answering them. Overall, I'm satisfied and can support the publication of the manuscript in the current form.

Reviewer #3:

Remarks to the Author:

Second Review of the manuscript "Optical-Field Driven Charge-Transfer Modulations near Composite Nanostructures" by Lee et al.

My opinion is that the paper has an interesting science. That is why I believe that it is worth to be published. It will make a valuable contribution to the field.

The authors worked hard to address the Referees' comments provided during the first round of reviews and, in many cases, this made the manuscript better.

Overall, I am reasonably satisfied with the authors' answers and revisions provided in response to my critique and suggestions (Referee 3). Thus, YES from me.

Answer to the reviewer's comments – October 2020

Reviewer #1 (Remarks to the Author):

I'm quite satisfied with the authors' response. The revised manuscript and supplemental materials are more insightful now. I do share some of the concerns of the Reviewer #2, but find that the authors did a great job answering them. Overall, I'm satisfied and can support the publication of the manuscript in the current form.

We are delighted to read that Reviewer #1 is satisfied with the answers we have provided. We would like to thank for his/her support and comments, which naturally helped improving the manuscript over the course of the reviewing process.

Reviewer #3 (Remarks to the Author):

Second Review of the manuscript “Optical-Field Driven Charge-Transfer Modulations near Composite Nanostructures” by Lee et al.

My opinion is that the paper has an interesting science. That is why I believe that it is worth to be published. It will make a valuable contribution to the field.

The authors worked hard to address the Referees' comments provided during the first round of reviews and, in many cases, this made the manuscript better.

Overall, I am reasonably satisfied with the authors' answers and revisions provided in response to my critique and suggestions (Referee 3). Thus, YES from me.

We are delighted to read that Reviewer #3 highlights the hard work we provided to address the Referees' comments and acknowledges the valuable contribution to the field our work will make. We would like to thank for his/her support and comments, which naturally helped improving the manuscript over the course of the reviewing process.